# Evaluation of Rapid Lateral-Flow Tests Directed against the SARS-CoV-2 Nucleoprotein Using Viral Suspensions Belonging to Different Lineages of SARS-CoV-2

**DOI:** 10.3390/v14122628

**Published:** 2022-11-25

**Authors:** Sylvie Pillet, Julien Courtieux, Sylvie Gonzalo, Issam Bechri, Thomas Bourlet, Martine Valette, Antonin Bal, Bruno Pozzetto

**Affiliations:** 1Department of Infectious Agents and Hygiene, University Hospital of Saint-Etienne, Avenue Albert Raimond, CEDEX 02, 42055 Saint-Etienne, France; 2Centre International de Recherche en Infectiologie (CIRI), 46 allée d’Italie, 69007 Lyon, France; 3Laboratory of Virology, Institute of Infectious Agents, University Hospital of Lyon, BP 2251, 69229 Lyon, France

**Keywords:** viral lineages of SARS-CoV-2, nucleocapsid protein, lateral-flow rapid test, cell culture, quantitative RT-PCR

## Abstract

Within the successive waves that occurred during the SARS-CoV-2 pandemic, recommendations arose to test symptomatic and contact subjects by using rapid antigen devices directed against the viral nucleocapsid protein with the aim to isolate contagious patients without delay. The objective of this study was to evaluate the ability of four rapid lateral-flow tests (RLFT) that were commercially available on the French market in 2022 to recognize various strains of SARS-CoV-2. Series of five-fold dilutions of seven viral suspensions belonging to different lineages of SARS-CoV-2 (19A, 20A, Alpha, Beta, Gamma, Delta and Omicron) were used to evaluate the analytical sensitivity of four commercially available RLFTs (manufacturers: Abbott, AAZ, Becton-Dickinson and Biospeedia). Cell culture and quantitative RT-PCR were used as references. Excellent correlations were observed for each lineage strain between the viral titer obtained via cell culture and the number of RNA copies measured by quantitative RT-PCR. Although the four tests were able to recognize all the tested variants, significant differences in terms of sensitivity were observed between the four RLFTs. Despite the limitation represented by the small number of devices and clinical isolates that were tested, this study contributed by rapidly comparing the sensitivity of SARS-CoV-2 RLFTs in the Omicron era.

## 1. Introduction

In the course of the pandemic of SARS-CoV-2 that emerged at the end of 2019 in Wuhan, China [1], several waves of infection occurred through time in different regions of the world. In 2020, the World Health Organization (WHO) identified different variants of concern (VOC) that replaced successively the original strain [2].

Despite the use of molecular tests that remain the reference method for SARS-CoV-2 detection in different clinical specimens and notably in the respiratory tract, more rapid tests based on the antigen detection of the conserved nucleocapsid region have been recommended for identifying and isolating infected patients without delay [3,4]. These rapid lateral-flow tests (RFLT) are based on the immuno-chromatographic method using proprietary monoclonal antibodies. A large number of RFLTs are commercially available worldwide; those authorized by the European Union (EU) have been registered on the EU common list with a specific device identification number [5]. With the successive emergence of the different VOC of SARS-CoV-2 over time, it is important to verify that these RFLTs are able to recognize the current circulating strains [6,7,8,9,10,11,12,13]. For instance, a team from the University Hospital of Geneva, Switzerland, investigated the ability of different antigen tests to detect the new variant strains, i.e., Delta [9] and the BA-1 subvariant of Omicron [13]. In July 2022, a Cochrane systematic review was published with the objective to assess the diagnostic accuracy of rapid point-of-care antigen tests for diagnosis of SARS-CoV-2 infection [14].

At the beginning of 2022, which established the predominance of the Omicron variant worldwide, the aim of this study was to evaluate the ability of four RLFTs that were commercially available on the French market in early 2022 to detect strains of different variants of SARS-CoV-2, including the Omicron one. Cell culture and true quantitative RT-PCR were used as references.

## 2. Materials and Methods

### 2.1. Viral Strains

The selected, different SARS-CoV-2 strains were sequenced and deposited at GISAID (https://www.gisaid.org/ accessed 20 November 2022) with the following accession numbers (Nextstrain/Pango classifications): EPI_ISL_1707038 (19A/B.38); EPI_ISL_1785075 (20A/B.1.160); EPI_ISL_1707039 Alpha variant (20I/B.1.1.7); EPI_ISL_768828 Beta variant (20H/B.1.351); EPI_ISL_1359892 Gamma variant (20J/P.1); EPI_ISL_1904989 Delta variant (21A/B.1.617.2); and EPI_ISL_7608613 Omicron variant (21K/BA.1). To make the reading easier, the different strains were named 19A, 20A, Alpha, Beta, Gamma, Delta and Omicron, respectively. The viral strains were kept frozen at −150 °C in a biosafety level 3 laboratory.

### 2.2. Titration of Viral Strains in Cell Culture

Cell cultures of each strain were performed in Dulbecco’s Modified Eagle Medium (DMEM, Fisher Scientific SAS, Illkirch, France) containing 2% embryonic calf serum and antibiotics; 150 µL of each dilution was transferred into 96-well microplates covered with Vero E6 cells (American Type Culture Collection (ATCC), CRL-1586, not authenticated but regularly tested for mycoplasma contamination). The plates were incubated at 37 °C in a 5% CO_2_ atmosphere. Infection efficiency was evaluated 5 days later by microscopic examination of the cytopathic effect. All experiments were performed in a biosafety level 3 laboratory. The viral titer was expressed in tissue culture infectious doses 50% (TCID_50_) per 200 µL by using the Reed and Muench formula [15].

### 2.3. Quantitative RT-PCR

The viral load of each five-fold dilution of viral suspension was determined using the QUANTI SARS-CoV-2 R-GENE^®^ kit (bioMérieux, Craponne, France) on an Applied Biosystems 7500 Fast after extraction of nucleic acids on the NUCLISENS^®^ easyMAG^®^ platform (bioMérieux, Craponne, France) as recommended by the manufacturer. The quantification kit was provided with 4 concentrations of a plasmid including a nucleocapsid-encoding gene fragment, allowing calculation of the viral load that was expressed as log_10_ copies of viral RNA per 200 µL of viral suspension.

### 2.4. Rapid Antigen Tests (RFLT)

The characteristics of the 4 RLFTs that were tested are shown in Table 1. The tests were performed according to each manufacturer’s instructions. Swabs furnished with each kit were immersed in the different five-fold viral suspensions of each of the different variants; the antigen content of the swab was then extracted in the amount of buffer recommended for each RLFT. The appropriate volume of the reactive medium was deposited onto the RLFT kit device. After the migration time recommended by each manufacturer (10 to 15 min), the reading was performed with naked eyes by two independent observers, except for the Veritor™ test, for which an optical reader was used, as recommended. The eye reading of each viral suspension was interpreted as positive, weak positive or negative, whereas the Veritor™ test was interpreted as positive or negative. All antigen testing was conducted under biosafety level 3 conditions using live virus.

### 2.5. Sequence and Statistical Analyses

Sequences of the nucleocapsid protein of each of the viral strains were taken from GISAID (see accession numbers above); their alignment was performed using the MUSCLE tool.

The correlation between cell culture data expressed as TCID_50_ per 200 µL and quantitative RT-PCR expressed as log_10_ copies of viral RNA per 200 µL of viral suspension was calculated by using the Pearson R coefficient of correlation with an alpha risk of 5%.

## 3. Results

Figure 1 illustrates the correlation observed for each lineage strain between viral titers obtained in cell culture and number of RNA copies obtained by quantitative RT-PCR. Excellent correlations were found, with respective R correlation coefficients of 0.982, 0.993, 0.962, 0.984, 0.998, 0.999 and 0.996 for lineages 19A, 20A (D614G), Alpha variant, Beta variant, Gamma variant, Delta variant and Omicron variant.

The performances of the four RLFTs are illustrated in Figure 2A regarding cell culture and in Figure 2B regarding quantitative RT-PCR.

Appendix A illustrates the correspondence between the cycle threshold (C_T_) values obtained for each set of data and the true viral loads quantified by the QUANTI SARS-CoV-2 R-GENE^®^ kit.

The four tests were able to recognize all the tested variants. The alignment of the sequences of the nucleocapsid (N) protein, which is the target of the four tests (Table 1), of the seven viral strains showed a few mutations in this protein, including D3L, R203K, G204R and S235F for the Alpha, Beta and Gamma variants; D63G, R203M, G275C and D377Y for the Delta variant; and P13L, deletion 31–33, R203K and G204R for the Omicron variant (Figure 3). These point mutations did not affect the ability of the different tests to recognize the N protein.

As shown in Table 2, the BioSpeedia test was the most sensitive one for six of the seven lineages; the Abbott test exhibited a very close sensitivity, with the best result for five of the seven lineages; the AAZ test exhibited the best result for three of the seven lineages, whereas the BD Veritor™ test was the last sensitive test for six of the seven lineages.

## 4. Discussion

The first point underlined by this study concerns the ability of the four RLFTs to recognize the different strains of SARS-CoV-2 that circulated worldwide from 2019 up to now despite minor mutations shown in the *N* gene (Figure 3). These results obtained with successive dilutions of cultured clinical isolates belonging to representative lineages of SARS-CoV-2 confirm those of previous studies based on respiratory specimens taken directly from patients [6,7,8,10,12,13]. Our approach, also proposed by other authors [9,11], limits the number of tested samples but allows the data to be extended to multiple lineages. Unsurprisingly, an excellent correlation was observed between the viral titers expressed in TCID_50_ in cell culture and the viral loads expressed in RNA copies in quantitative RT-PCR (Figure 1). Incidentally, this finding supports the use of viral loads obtained by quantitative molecular tests as an equivalent of cell culturability (see Appendix A) that, as of now, represents the best proxy of a patient’s infectiousness in clinical practice [16,17].

Another lesson driven from this study is the variability of the analytical sensitivity of the different RLFTs available on the market. The PanBio test, already clinically evaluated in multiples studies (Table 1 and [6,9,10,11,18]) and the BioSpeedia test [19,20] had better sensitivity towards the different SARS-CoV-2 isolates; the AAZ test [21] exhibited a close sensitivity. By contrast, the Veritor test, despite the availability of a reader that could optimize the detection of positive bands, showed a lower sensitivity towards most of the strains (Figure 2 and Table 2). These results are in accordance with those obtained in several clinical studies in which the performance of this device was compared to similar, other ones [6,8,12,22].

A recent Cochrane review dedicated to rapid antigen tests compiled the results of 155 cohort studies [14]; it confirmed that the specificity of these devices is good but that their sensitivity can vary considerably from one kit to another. This sensitivity was mostly related to the presence or not of symptoms and to the time at which the test was performed with regards to the beginning of symptoms for symptomatic patients. The average sensitivity was higher in the first week after symptom onset than in the second week of symptoms (see also [16]). For those who were asymptomatic at the time of testing, sensitivity was higher when an epidemiological exposure to SARS-CoV-2 was suspected compared to blind screening of large populations (see also [20]). Average sensitivities by brand ranged from 34.3% to 91.3% in symptomatic participants and from 28.6% to 77.8% in asymptomatic ones [14]. Even if we agree that the more accurate way to evaluate the clinical performance of a rapid test is to perform a field cohort study, the use of clinical strains obtained by cell culture, as proposed by our study and by others [9,11], represents a convenient complementary approach for comparing rapidly the sensitivities of different RLFT kits, which may be very different from those announced in the package inserts of each test (see Table 1).

In conclusion, despite the limitations due to the small number of devices and clinical isolates that were tested, this study was helpful in comparing the sensitivities of SARS-CoV-2 rapid antigen tests in the Omicron era.

## Figures and Tables

**Figure 1 viruses-14-02628-f001:**
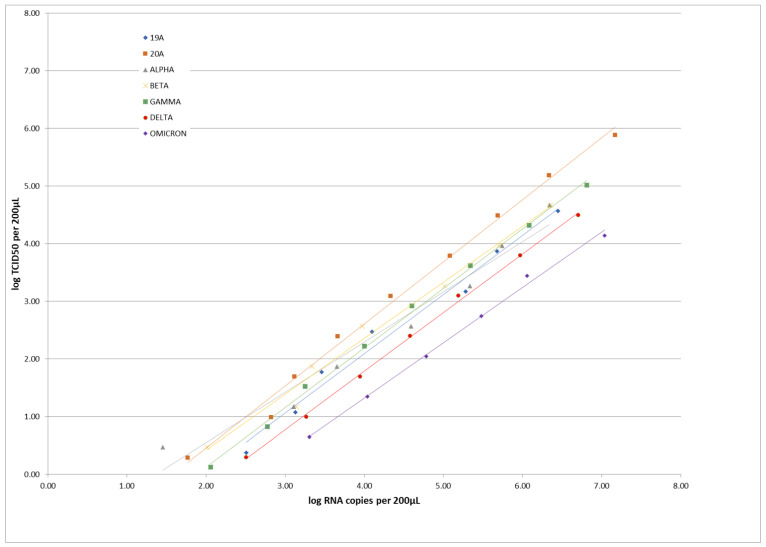
Correlation of the viral loads expressed as log_10_ copies/200 µL and log_10_ TCID_50_/200 µL for the 5-fold dilutions of the seven tested strains belonging to different lineages of SARS-CoV-2.

**Figure 2 viruses-14-02628-f002:**
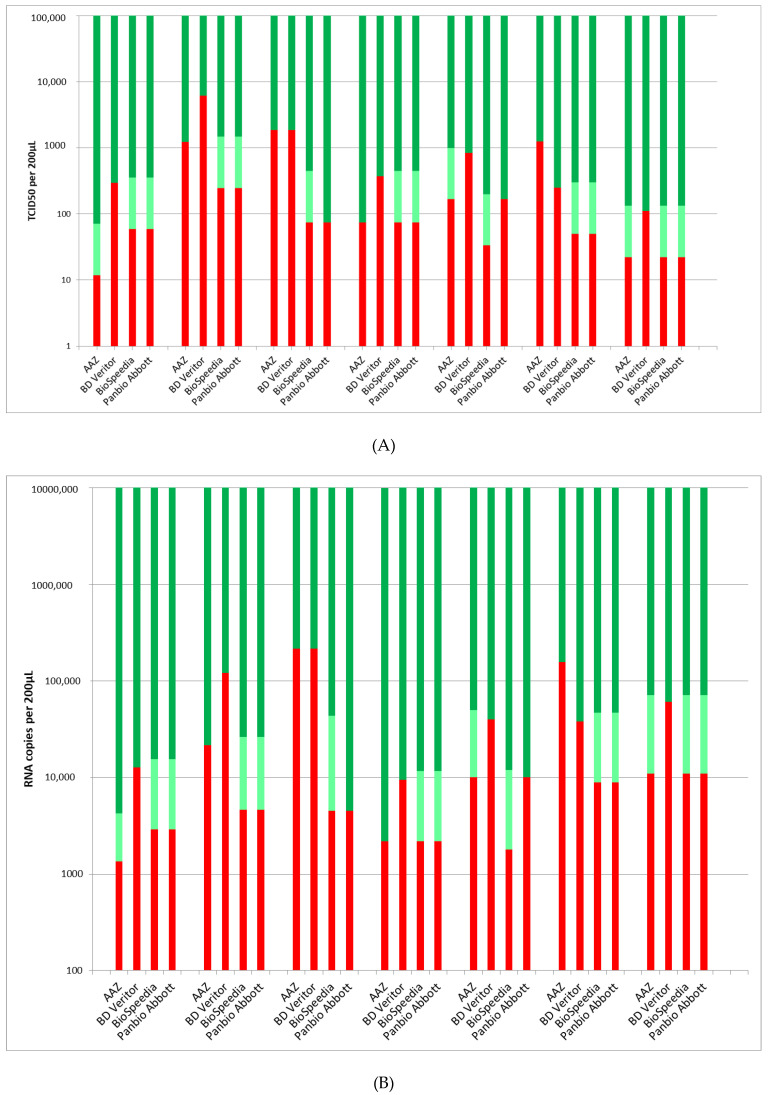
Evaluation of the sensitivity of the 4 tested RFLTs regarding the different SARS-CoV-2 lineages by comparison to the viral titer (expressed in TCID_50_) in cell culture (**A**) and to viral load (expressed in RNA copies) in quantitative RT-PCR (**B**). Color symbols: dark green = positive; light green = weakly positive; red = negative.

**Figure 3 viruses-14-02628-f003:**
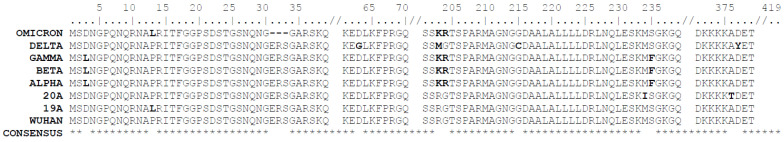
Alignment of the amino acid sequences of the *N* gene of the 7 viral strains tested in this study by comparison to that of the Wuhan strain retrieved for GISAID and used as reference. The accession numbers of the stains in GISAID are given in the text. The MUSCLE tool was used to perform the alignment. Only mutated regions are shown on the figure (black letters). Note the deleted sequence of three amino acids in position 31–33 for the Omicron variant.

**Table 1 viruses-14-02628-t001:** Characteristics of the four tested RLFTs, which were all registered on the European Union (EU) common list [5].

Manufacturer	Commercial Name	Device Identification Number	Clinical Performance (Independent Results) (Tested Specimens Were NP If Unspecified)	Clinical Performance (Manufacturer)	Countries of Completed Validation Studies	SARS-CoV-2 Target Protein	Specimen	Date of Inclusion in EU Common List
AAZ-LMB	COVID-Viro	1833	** *Prospective clinical field study* ** − France: N = 76; ss: 94.7%, sp: 100%	Nasal swab, NP swab ss: 96.6%; sp: 100%	France Switzerland	Nucleoprotein	Nasal swab NP swab	10 May 2021
Abbott Rapid Diagnostics	Panbio™ COVID-19 Ag Rapid Test	1232	** *Prospective clinical field studies* ** − Belgium: N = 57; ss: 79%; sp 100%. − Netherlands: N = 1367 in Utrecht and N = 208 in Aruba; ss: 72.6% in Utrecht and 81.0% in Aruba; sp: 100% in both settings − Portugal: N = 83; ss: 63%; sp: 100% − Sweden: N = 245 (specimens non specified); ss: 59%, sp: 100% ** *FIND evaluation studies* ** − Germany: N = 1108; ss: 90.8%; sp: 99.9% − Switzerland: N = 535; ss: 85.6%; sp: 100% − India: N = 526; ss: 61.3–100%; sp: 100%	NP swab (C_T_ ≤ 33) ss: 91.4%; sp: 99.8% Nasal swab (C_T_ ≤ 33) ss: 98.1%; sp: 99.8%	Belgium Germany (2) Spain Finland Netherlands (5) Portugal Sweden Switzerland India Norway UK	Nucleoprotein	Nasal swab NP swab	17 February 2021
Becton-Dickinson	BD Veritor™ System for Rapid Detection of SARS-CoV-2	1065	** *Prospective clinical field studies* ** − Spain: N = 476 (nasal swab); ss: 92%; sp: 98.6%. − Netherlands: N = 979 (nasal mid-turbinate + oro-pharyngeal swab); ss: 79.5%; sp: 99.8% − Sweden: N = 245 (specimens non specified); ss: 45%; sp: 97%	Nasal swab ss: 91.1 % sp: 99.6 %	Germany (2) Spain Netherlands Sweden	Nucleoprotein	Nasal swab	7 July 2021
BioSpeedia International	COVID19 Speed-Antigen Test BSD_0503	2380	** *Prospective clinical field study* ** − France: N = 620; ss: 95.29%; sp: 99.73%	NP swab ss: 97.5% sp: 99.3%	France	Nucleocapsid protein	NP swab	21 January 2022

Abbreviations: NP: nasopharyngeal; EU: European Union; N: number of tested specimens; ss: sensitivity; sp: specificity; CT: cycle threshold; FIND: Foundation for Innovative New Diagnostics.

**Table 2 viruses-14-02628-t002:** Comparison of the analytical sensitivity of the four tested RFLTs regarding different SARS-CoV-2 lineages.

Tested Strains	Sensitivity Ranking
19A	AAZ > BS = Ab > BD
20A	BS = Ab > AAZ > BD
Alpha	BS = Ab > AAZ = BD
Beta	BS = Ab = AAZ > BD
Gamma	BS > AAZ = Ab > BD
Delta	BS = Ab > BD > AAZ
Omicron	BS = Ab = AAZ > BD

Ab: Panbio™ COVID-19 Ag Rapid Test, Abbott Rapid Diagnostics. AAZ: Covid-Viro, AAZ-LMB. BS: COVID19 Speed-Antigen Test, BioSpeedia International. BD: BD Veritor™ System for Rapid Detection of SARS-CoV-2, Becton Dickinson.

## Data Availability

Full data are available on request to the corresponding author.

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
