# Peer review of "Evaluation of Rapid Lateral-Flow Tests Directed against the SARS-CoV-2 Nucleoprotein Using Viral Suspensions Belonging to Different Lineages of SARS-CoV-2"

_viruses, 2022, doi:10.3390/v14122628_

Round 1

Reviewer 1 Report

This is a useful and interesting paper. The author have tested four different rapid antigen tests for their ability to detect SARS-CoV-2 in samples prepared from isolates. Dilutions of viral suspensions from different variants were analyzed by RLFT, PCR and cell culture. The authors conclude that all four commercial tests detect the newer variants, despite mutations in the gene of the nucleocapsid protein detected, and that there are differences in sensitivity.

The strength of the paper is that the study was well designed and performed; the weaknesses include a low number of samples and a modest level of novelty, as well as a qualitative read-out of the RLFT results.

The following issues should be addressed.

- The text should be carefully checked and edited. For example, there are left-over instructions from the template in the text ("This section may be divided by subheadings. It should provide a concise and precise description of the experimental results, their interpretation, as well as the experimental conclusions that can be drawn.")! For Table 1, it is unclear whether this is copy-paste from an official document or a table that was compiled by the authors. The reference link does not work, and there are still many spellchecker marks in the table.

- This reviewer would have enjoyed also seeing the viral loads expressed in ct values, or a comparison with ct values in the Discussion.

- The Introduction is unusually brief, and so are other parts. The Introduction should discuss what other publications exist on the topic.

- Self praise should be avoided ("this study represents an original contribution ..", found more than once).

- The manuscript requires language-polishing, e.g.

correlations performed -> obtained

At the era -> In the era

immerged into -> immersed in

punctual changes -> point mutations

more sensitive results -> better sensitivity

etc.

Reviewer 2 Report

Abstract; suggest using “Series of five-fold dilutions” 

Mixing WHO and Nextstrain nomenclature of the variants, maybe use Nextstrain or Pangolin naming conventions.

Page 2, line 46; “at the omicron era” can you be more specific?

The results shown in this paper are in line with what would be expected, the sensitivity is dependent on the amount of antigen and the affinity/avidity of the detection antibodies.

The major contributor to errors using lateral flow kits is the user and sample collection, not the kit.

In a clinical setting, the most interesting information about diagnostics kits is to define sensitivity, specificity, precision, and accuracy.

Round 2

Reviewer 2 Report

none